# Chromosome Changes in Soma and Germ Line: Heritability and Evolutionary Outcome

**DOI:** 10.3390/genes13040602

**Published:** 2022-03-28

**Authors:** Irina Bakloushinskaya

**Affiliations:** Koltzov Institute of Developmental Biology, Russian Academy of Sciences, 119334 Moscow, Russia; i.bakloushinskaya@idbras.ru

**Keywords:** chromosome changes, speciation, Robertsonian translocations, meiotic checkpoints, mitotic checkpoints, chromothripsis, spermatogenesis, oogenesis

## Abstract

The origin and inheritance of chromosome changes provide the essential foundation for natural selection and evolution. The evolutionary fate of chromosome changes depends on the place and time of their emergence and is controlled by checkpoints in mitosis and meiosis. Estimating whether the altered genome can be passed to subsequent generations should be central when we consider a particular genome rearrangement. Through comparative analysis of chromosome rearrangements in soma and germ line, the potential impact of macromutations such as chromothripsis or chromoplexy appears to be fascinating. What happens with chromosomes during the early development, and which alterations lead to mosaicism are other poorly studied but undoubtedly essential issues. The evolutionary impact can be gained most effectively through chromosome rearrangements arising in male meiosis I and in female meiosis II, which are the last divisions following fertilization. The diversity of genome organization has unique features in distinct animals; the chromosome changes, their internal relations, and some factors safeguarding genome maintenance in generations under natural selection were considered for mammals.

## 1. Introduction

Our understanding of chromosomes and their essence, structure, and function has been dramatically transformed throughout the research history, from first pictures drawn by F. Nägeli [1] and his hypotheses on the hereditary role of chromosomes [2,3] to the present day, when chromosomes are considered as carriers of genetic information and guardians of heredity. Maintenance and transmission of genetic information to descendants are essential for living beings. This simple statement conceals a plethora of problems. The genome must be stable and variable at the same time; that is, it must have a structure and mechanisms to preserve basic information and generate changes, which might be helpful in a modifying environment. In other words, it is important to understand how to maintain the species-specific genome, which is crucial for a healthy being, and obtain new features for evolution. Error-free repair of DNA breaks and replication of DNA provide the basics for proper mechanisms to preserve, copy, and transmit DNA packaged to chromosomes from cell to cell and from parents to offspring. Nevertheless, these processes are not perfectly consistent.

De novo single nucleotide variants (SNVs), inversions, insertion/deletions (indels), de novo copy number variants, and aneuploidies arise in the preimplantation embryo, and later, in the soma and germ line [4,5]. The term “chromoanagenesis” covers at least three processes, chromothripsis, chromoanasynthesis, and chromoplexy, which were first detected in distinct cancers [6,7,8]. Modeling of these processes revealed that chromosomes undergo fragmentation and consequent fusions together with deletions, emergence of micronuclei consisting of fragmented parts of chromosomes, and anaphase bridges complicating the transmission of changed chromosomes to descendant cells [9].

At least four mechanisms have the potential to generate genome rearrangements: non-allelic homologous recombination (NAHR), erroneous repair after double-strand DNA breaks, non-homologous end-joining (NHEJ), replication errors, and retrotransposition [10,11,12]. Distinct mechanisms, such as NAHR and NHEJ, or DNA replication errors are likely to produce the same types of rearrangements: inversions, deletions, duplications, and translocations. Various de novo genomic rearrangements can emerge during gametogenesis [13,14]. Since gametogenesis includes mitosis and meiosis, potential rearrangements differ depending on these stages. NAHR is a specific meiotic mechanism; erroneous repair after double-strand DNA breaks might also occur during mitosis. DNA single-strand breaks might be a source of chromosome rearrangements in the case of downregulation of homologous recombination factors expression [15]. All mentioned rearrangements are classified as structural variations (SVs) and possibly inherited in cell lines or descendants if they pass mitotic/meiotic checkpoints. SVs are involved in genomic disorders, congenital diseases, and cancers. At least a half of changes during cancer genesis are whole chromosomes or chromosome arms gains and losses; a variety of chromosome set rearrangements can arise due to cytokinesis failure or endoreduplication [16,17].

The three-dimensional structure of chromosomes and their positioning in the nuclei are essential for genome functioning and faithful transmission to descendants. Single-cell analysis and high-resolution microscopy offer an opportunity to see three-dimensional genome structuring as static pictures, which should match the dynamics of chromosome organization [18,19,20,21]. Many studies have examined the conservative evolutionary arrangements of chromosome territory [22] alongside species-specific changes, which probably lead to reproductive isolation [23,24]. In addition to duplications, insertions, and deletions, the occurrence of which changes the DNA structure, chromosomes undergo large-scale structural changes that have little effect on nucleotide sequences (although microdeletions are probable) but significantly alter the chromosome and nuclear structure, as in Robertsonian translocations (Rbs) or Whole Arms Translocations (WARTs). The patterning of chromatin domains and compartments appeared to be highly variable in different cell types [25,26]. TADs undergo total rearrangements in the cell cycle; these functional reorganizations are essential for the transcription dynamics at different times and places during lifespan [27,28,29]. Chromosome changes, such as duplication, indels, inversions, or translocations that destroy TAD architecture, might cause developmental defects in mammals due to incorrect enhancer–promoter interactions or other alterations [30].

In addition to the apparent effects on genome function, chromosome rearrangements resulting in their location changes in the nucleus also lead to problems of synapsis in the meiosis, recombination failure, and complete or partial reproductive isolation [24].

The concept of natural selection as applied to genome evolution makes it possible to estimate the genetic variability accumulated during an individual’s life as significant for the individual’s own (i.e., adaptive) or his/her descendants in a case of lagged adaptiveness. As a rule, genome variability in somatic cells belongs to the first pool, and in germ cells to the second one. Some mechanisms contribute to both, and an exception is early embryonic development preceding the formation of primordial germ cells (PGCs). This paper aims to draw attention to evolutionarily significant chromosome changes and highlight the place and time they might arise.

## 2. Chromosome Rearrangements in Early Development

The type and prevalence of de novo mutations appear to be different throughout development stages [31,32]. The evolutionary consequences of de novo mutations are associated with their essence and the time and place they originated.

### 2.1. Start of a New Life

A start of new life in multicellular diploid organisms, such as mammals, might be treated as restoring the diploid state in the zygote by joining two haploid gametes. The fusion of male and female gametes triggers the second meiosis division in the oocyte, resulting in the second polar body and two pronuclei with maternal and paternal haploid sets. The assumption would be that then, the two pronuclei merge and the chromosomes mix, which means that there should be no separation of the parental genomes during the first division. However, it was demonstrated that several events should take place for proper development. First, a sperm centrosome should duplicate, then two spindles develop, and dynein adjusts chromosomes toward centrosomes; the main issue is to cluster and separate parental genomes, as clustering alterations lead to aneuploidy and micronuclei formation, a relatively frequent phenomenon [33,34,35].

The first stages of genome activation development in maternal to zygotic transition are pivotal because the zygotic genome undergoes restructuring for further activation [36]. The higher-order chromatin folding processes in maternal to zygotic transition reflect the need for checkpoints. It is known that maternal RNA degradation is necessary for switching on the zygotic genome; many epigenetic mechanisms are involved in these processes. The genetic program will not be carried out if the chromatin architecture is not correctly established at the beginning of embryonic development [37]. In such a case, embryonic development will terminate.

### 2.2. Early Development, Alterations, and Mosaicism

The notion of genome rearrangements during early development is a highly complicated issue for study. Undoubtedly, any errors during the first divisions can lead to eliminating the embryo or the appearance of mosaics. Mosaicism is a common feature of mammals, including humans, starting from the earliest cleavages and becoming more evident in aging genomes [38]. Mosaicism might be a result of the nonequilibrium division of different blastomeres or, later during the lifespan, the existence of various cell clones. However, it could happen that only one cell will become the progenitor of a clone, leading to somatic mosaicism [39]. Loss of X or Y chromosomes is common in somatic cells in the aged genome in humans [40,41,42,43].

Another kind of chromosome changes, the emergence of a dicentric Y chromosome, leads to sterility in humans [44]. Supposedly, this rearrangement occurs early in development, probably as a result of recombination between sister chromatids or as a result of a break or deletion followed by a delayed joining of sister chromatids. These mechanisms may contribute to various mosaicism disorders, such as cell lines without a Y chromosome in sterile males 45,X [45]. Numerous variants of dicentric Rbs were shown in cattle, buffaloes, sheep, and goats [46]. Even though different variants are formed, such chromosomes, probably due to the presence of two centromeres, are subjected to selection in the form of centromeric drive and are eliminated in the lineage within several generations.

Such aberrations have no evolutionary output in most cases because they are not passed to gametes. However, if mosaicism occurs in the germ line, the consequences should be significant due to different genetic values of the gametes; moreover, the problem of mosaicism is extensively debated for humans, with recent awareness that any manifestations of mosaicism must be considered, as the health consequences can be severe [47].

The separation of somatic and germ cells in mammalian development was a longstanding quest for generations of embryologists [48,49,50]. Molecular mechanisms for PGCs specification in the pre-gastrulation period are not conserved for mammals, and this could mean that there are different mechanisms for maintaining genetic stability [51,52]. If chromosome changes emerge during the PGCs formation and do not dramatically affect gametogenesis and fertility, the changes have a chance to be passed to descendants and probably possess evolutionary significance [53].

## 3. Gametogenesis

Mitosis and meiosis start from the duplication of chromosomes and other cellular structures; the next step is the packaging of chromosomes, alignment, and the precise separation into two daughter cells (mitosis), four spermatids (male meiosis), or an oocyte and two polar bodies (female meiosis). Differences between sexes in the meiosis and its products determine the meiotic drive as one of natural selection’s mechanisms [54]. All steps of gametogenesis should be strictly controlled for proper development.

### 3.1. Mitotic and Meiotic Checkpoints

Checkpoint mechanisms (specific signaling pathways) must recognize the accuracy of events taking place and allow the process to go forward [55,56]. If something is going wrong, the checkpoints stop the progression of the cell cycle and activate repair mechanisms. Checkpoints de facto are natural selection mechanisms. In mitosis and meiosis (male and female), the checkpoints for selecting mutations might be similar or different depending on the stage [57].

The spindle assembly checkpoints (SAC) operate in mitosis and meiosis to assess the attachment of each chromosome to the spindle. A failure to pass this control leads to a halt of cell divisions before the anaphase [58,59]. The molecular mechanisms of spindle assembly checkpoints were reviewed in [60]. SAC prevents mis-segregation leading to chromosomal instability and aneuploidy, which might be catastrophic for the cell [61]. Many recent studies show that cancer cells are altered mainly in their ability to exit the cell cycle and therefore continue to divide rather than undergo uncontrolled cell division [62]. Altered segregation produces karyotypic heterogeneity in tumoral subclones and leads to cancer progression [63,64].

Meiotic checkpoint machinery mainly works in the prophase I and provides recombination and synapsis checkpoints, the most important for proper meiotic progression. Homologous pairing, synapsis, and recombination are safeguarded by specific checkpoint proteins ATM/ATR (ATM (ataxia telangiectasia mutated) and ATR (ATM and Rad3-related)), CHK1, CHK2, HORMA, TRIP13, and others [65,66,67,68]. Meiotic silencing of unsynapsed chromatin (MSUC) develops by recruiting ATR via HORMAD1/2. ATR accelerates phosphorylation of H2AFX, which expands into the chromatin loops to recruit silencing factors, e.g., g-H2AFX-binding factor MDC1 [69], and to initiate the heterochromatinization and MSUC. Numerous kinases, including cyclin-dependent kinases (Cdks), participate in controlling cell cycle progression. The inhibition of CDK1 can halt mitosis, and CDK2 is involved in meiosis regulation [70,71].

The other checkpoints are equally important, such as detecting DSBs and providing a DNA damage response in which the roles of ATM and ATR appear similar between mitosis and meiosis. DNA damage checkpoint proteins controlling meiotic prophase events are evolutionarily conserved [72,73]. Recombination and homologous chromosome pairing are controlled by checkpoints in male meiosis more effectively, and oocytes are more tolerant to alterations [74,75]. The appropriate chromosomal segregation is probably controlled in a sexually dimorphic way due to differences in some proteins, such as ZMYM3 (a member of the MYM-type zinc finger protein family and a component of an LSD1-containing transcriptional repressor complex) in mice [76]. The study of compromised SAC protein function in meiosis is rather difficult and results are uncertain [77].

In female meiosis, the DNA damage checkpoint can eliminate oocytes with chromosome synapsis failure [78,79,80]. Non-growing oocytes arrested at the prophase I efficiently use homological recombination for DNA DSB repair maintenance of genetic integrity in oocytes [81]. Double-strand breaks in the fully grown mouse oocytes may induce short-scale DNA replication and damage amplification, resulting in complex genomic rearrangements [82].

### 3.2. Chromothripsis in Germ Line

Chromothripsis was observed during spermatogenesis (premeiotic division of diploid spermatogonia, spermiogenesis) (Figure 1) and in first cleavages (preimplantation embryos) but was not detected during oogenesis [83,84]. DNA repair capacity is relatively low in the male germ line compared to somatic cells; moreover, the mutation rate is higher in spermatogenesis than in oogenesis [85]. The extent of rearrangements can be significant [86], which, undoubtedly, should have an evolutionary consequence. Probably, the massive rearrangements (genome reshuffling) described in rodents [87,88] can be explained by the acts of chromoanagenesis in the germ lines.

### 3.3. Female Meiotic Drive as an Evolutionary Force

The significant differences between male and female meiosis manifest in the timing of events and the products. In males, meiosis culminates in four haploid spermatids that differentiate into four spermatozoa (Figure 1). In females, the cytokinesis is asymmetric, resulting in only one functional oocyte and two polar bodies being eliminated (Figure 1). The disparity of the first and second divisions of meiosis determines the input of selection at these stages in the formation of the final haploid gamete, an egg. The first division results in the formation of the oocyte II and the first polar body, which have a haploid set of chromosomes but doubled DNA content (1n, 2C). This process occurs after synapsis and chromosome recombination, so the oocyte II and the polar body genomes tend to be distinct. Two dissimilar genomes can also be obtained during the second division, although with 1n, 1C, since another polar body and the egg itself are formed. In fact, the polar bodies become a reservoir for unfavored karyotypes, which leads to rapid changes in the chromosome set, making the meiotic drive an evolutionary force [89,90].

Studying the transmission ratio distortion in offspring of heterozygous female carriers of Robertsonian translocations (Rbs) [91] reinforced the basics of the female meiotic drive as the main engine for chromosome speciation, especially in a case of partial monobrachial homology of certain chromosomes [92,93,94].

## 4. Chromosomal Speciation

The diversity of structural and numerical characteristics of mammalian chromosomal sets is very high; diploid numbers vary from 2n = 6 in *Muntiacus muntjak vaginalis* to 2n = 102 in *Tympanoctomys barrerae* [95,96]. However, despite longstanding recognition of the karyotype as a species-specific characteristic, the causality of chromosome rearrangements (CRs) for speciation is still disputable unless there is an extended history of the conception of “chromosomal speciation” [97,98,99]. Reduced gene flow in rearranged chromosome segments leads to divergence and speciation. The main question remains whether rearrangements are the cause of diversification, or whether forms that have diversified for other reasons are gradually obtaining rearrangements. Darwin’s theory [100] implies that changes accumulate gradually. “Modern Synthesis” [101,102] incorporated a genetic basis and expanded these assumptions to the genome. Massive rearrangements (e.g., chromoanagenesis) can lead to radical changes in the genome and provide a rationale for moving beyond the dominant evolutionary significance of gradual change to a punctuated equilibrium assessment [103,104]. The rationale on whether chromosome rearrangements are the main causative is consistent with Gould’s concept because the main disagreement concerns whether rearrangements are the starting point for divergence and the idea that evolution can occur in distinct modes, not only gradually but with very rapid events of speciation, as in the punctuated equilibria concept.

Massive complex rearrangements were recently named catastrophic events or genome chaos and discussed within the concept of macroevolution in cancer studies [105,106]. However, in the Modern Synthesis, macroevolution was defined as evolution above the species level, leading to the origin of large taxa [107]. The term “macromutation” was introduced in the concept of “hopeful monsters”, i.e., rapid and sudden speciation appeared as morphological transformation by R. Goldschmidt [108]; in this framework, the evolutionary significance of drastic changes in the genome is apparent. The term “macromutation” applied to massive chromosome changes seems more suitable than “macroevolution”, which dominates in cancer studies [109,110]. The theory of cancer cannot be regarded as an evolutionary one from the classic point of view since evolution means the emergence of new forms or species [111]. The realized possibility of reproducing, i.e., passing the genome to descendants, is the fundamental success criterion and selection indicator. Obviously, cancer development is opposite to heritability. The example of the Tasmanian devil, suffering from the facial tumor disease, in which massive changes in chromosomes were observed and became the cause of extinction, confirms this point [112].

Genomic chaos studies made a theoretically significant breakthrough, reasoning that natural selection operates at all levels of genome organization [113,114]. This idea was stated much earlier by a famous cytogeneticist, C. Darlington: “selection therefore acts on the genetic system at every level, gene or chromosome, cell and individual, and in every stage and process, haploid and diploid, mitotic and meiotic, embryonic and adult” [115] (pp. 130–131). The internal relations and certain factors ensuring genome maintenance should undoubtedly be sustained by natural selection.

The term “chromosomics” was recently introduced for joining cytogenetics with genomics [116]. Profound results were achieved by genome sequencing in model species, in house mice [117], and in *Daphnia* [118]. The involvement in speciation has been shown for inversions since the classic works on *Drosophila* [119]. Reciprocal translocations have been described for different phyla and represent a diverse pool causing a large part of the variability in humans, including germ line mutations [120,121,122,123]. Mutations of this kind alter the linkage groups and can be the source of changes in regulatory sequences, leading to various developmental abnormalities and diseases. The impact of such chromosome structure alterations on genome functioning is relatively straightforward. Heterologous synapsis can lead to suppressed recombination, sterility, or heterologous recombination [124,125]. The case of large-block rearrangements when chromosomes undergo fusions (Rbs), fissions, or arm exchanges (Whole Arms Translocations, WARTs) turns out to be less clear. Unless massive changes in linkage groups are not expected in this case, and synapsis has a chance to be homologous, the chromosomal fusions impact the recombination and alter the organization of topologically associating domains (TADs) [126].

Mutation bias was recently demonstrated for *Arabidopsis thaliana* [127], in which SNVs and indels were less frequent in gene bodies and essential genes in the soma and germ line mutations. Recently, a comprehensive analysis of chromosome-scale genome sequences analysis claimed the existence of chromosome-scale conservation of synteny in lower metazoans and revealed three basic types of chromosome rearrangements in the evolution: Robertsonian translocations (and end-to-end fusion), centric insertions, and a new type, fusion-with-mixing, similar in outcome to chromothripsis [128].

### 4.1. Robertsonian Translocations and WARTs

Robertsonian translocations involve whole arms of chromosomes; in the human genome, acrocentrics 13, 14, 15, 21, and 22 are most prone to these rearrangements. The rate of the Rbs in the human population is approximately 1 in 800, the highest for chromosome rearrangements [129]. Rbs are the most common variant of chromosome changes in mammals [98].

In cattle, a specific monocentric Rb between BTA1 and BTA29 is widespread; this single translocation has minimal impact on animal viability and fertility. Apparently, it arises in different breeding lines; furthermore, this translocation was also shown in the ancestral form, which indicates the karyotype prone to such a rearrangement [130,131,132]. In cases of partial or monobrachial homology, or in cases of numerous Rbs, the synapsis might be prolonged, leading to meiotic failure in some cells and decreasing fertility [133,134,135,136]. Examples of bats, shrews, mole voles, mice, and rock-wallabies demonstrate the restriction of the gene flow due to monobrachial homology, presumably resulting in speciation [137,138,139,140,141,142,143]. In mice, numerous variants, probably originated by WARTs, possess a complicated hybridization system through parental form [144]. WARTs have been described in the same groups with Rbs, which may evidence a common mechanism of origin for these rearrangements. In some rare cases, the numerous rearrangements, including tandem fusion, create a burst of intraspecific variability, as in Evoron vole [145].

Genome instability may lead to extinction or rapid speciation. The possibility of the emergence of chromosome rearrangements in germ line removes one of the most acute questions concerning the expansion and fixation of rearrangements in the population. A large number of altered gametes in spermatogenesis or offspring from a single female in case of changes in oogenesis enable rearrangements to fix in a population in several generations. Modeling predicts that a slight decrease in viability/fertility would allow obtaining divergent lines even in the case of sympatry [146].

### 4.2. Looking for a Model of Robertsonian Translocations Origin

Several models provide possible mechanisms of Robertsonian translocations formation by assessing spatial arrangements of chromosomes in the nucleus, their attachment to the nuclear envelope, and movements during the cell cycle [147,148].

One of the earliest hypotheses that suggested telomere loss or inactivation [149] received no support in mice because telomere shortening in germ line cells had a deleterious effect, so the efforts were turned to specific centromeric minor satellite sequences [150,151].

The concept of centromeric drive incorporates essentially unrelated mechanisms, referred to as “true” (distinct in male and female) and “killer” meiotic drive (known mostly in male meiosis) [152]. “Killer” meiotic drive appears due to specific alleles, for example, the *t*-haplotype in mouse lines, which makes viable males sterile. The system demonstrates lagged inadaptation and stresses the main point of evolution—the ability to reproduce.

The concept of “true” centromere drive supposes the emergence of “strong” and “weak” centromeres and the promotion of preferential transmission for chromosomes with “stronger” centromeres into eggs due to spindle asymmetry in the first meiotic division in mice [153,154]. Recently, some requirements for centromere drive were demonstrated [155]. The meiotic I spindle should be asymmetric; the CDC42 (cell division control protein 42) signaling from the oocyte cortex regulates the microtubule tyrosination and creates the asymmetric spindle [156]. Another important dimensional preference for “strong” centromeres is positioning away from the cortex for being included in the egg. Nevertheless, it is still unclear if centromere drive promotes the rapid evolution of changed chromosome sets.

The main difference in numerous models concerning the origin of chromosome rearrangements during interphase is whether chromosome contact or DNA breaks occur first; the inheritance of rearrangements in generations has not been explained. The evolutionary integrative breakage model [157] paid attention to the origin and genomic distribution of evolutionary breakpoints. The integrative breakage model stands out on the assumption that chromosome rearrangements occur in the interphase nucleus, whereas the centromeric drive model explores specifics of chromosome separation in cell division. The last model does not provide the mechanism of the emergence of chromosomes with different numbers of arms or dicentrics; it focuses on the preferable transmission of altered chromosomes and might be compared with the meiotic drive concept.

We also proposed a model to explain the formation and inheritance of Robertsonian translocations [158]. Based on our study of rapidly evolved mole vole *Ellobius alaicus*, we hypothesized that one of the possible mechanisms for arising the Robertsonian translocations might be a change in chromosome behavior in meiosis. Contacts of non-homologous chromosomes might lead to the fusion with the formation of dicentric chromosomes in prophase I. In the proposed model, contact of chromosomes in meiosis is the first and most crucial event for forming the Robertsonian translocations, so the model was called “contact first in meiosis”.

Whether the formation of WARTs through chromoplexy is possible is still unknown. A proposed mechanism for chromoplexy implies that the process of multiple DSBs by a single catastrophic event occurs in the interphase at the transcriptional cluster. It can be assumed that the formation of transcription centers as a hub is a key to the replacement of the arms in dicentric biarmed chromosomes even if one of the centromeres is temporarily inactive. Since all experimental data were obtained for bone and soft tissue tumors, it is difficult to extrapolate these results to other cases [159].

To date, the most reasonable concepts seem to be those which pay attention to the mutual arrangement of chromosomes in the nucleus—the integrative breakage model [157], their attachment to the nuclear envelope [149], and patterns of attachment to the fission spindle, which undoubtedly correlate with the checkpoints of SACs, as well [156]. The models that integrate the mechanism of rearrangement inheritance possess the highest prognostic efficiency. Therefore, the model “contact first in meiosis” appeared to be the most easily incorporated into the theory of chromosomal speciation in the case of the Robertsonian translocations or WARTs.

## 5. Conclusions

Chromosomal rearrangements can occur throughout the lifespan in interphase nuclei, during mitosis or meiosis. Various processes, from single- and double-stranded DNA breaks, non-homologous recombination, chromosome fusion to massive rearrangements such as chromothripsis and, possibly, chromoplexy, result in changes in the structure of individual chromosomes and the chromosome set. Changes in chromosome structure lead to altered genome functioning and the emergence of species-specific features. The most significant contribution in changing the species gene pool comes from those rearrangements that arise in the cells participating in gamete generation, including PGCs [160,161]. Rearrangements arising during zygotic cleavage and early embryonic development provide significant input to genetic variation, as well as germ line de novo mutations at pre-meiotic stages and all stages of meiosis [162], including the culminating moment—fertilization, which not only gives rise to new life, but also due to meiotic drive determines the structure of the chromosome set of a zygote.

The prevalence of the rearrangements arising during gametogenesis is reasonable for evolutionary perspectives because it releases constraints upon the reproductive rate as well as the viability of the parental organisms. Therefore, the rearrangements arising in meiosis appear to be the most significant for the evolutionary perspective. Chromosome rearrangements occurring in somatic cells have no chance to be passed on to descendants but turn out to be causative for various diseases, including cancer, and, though lacking an evolutionary output, can be considered with some limitations in modeling the evolutionary process.

## Figures and Tables

**Figure 1 genes-13-00602-f001:**
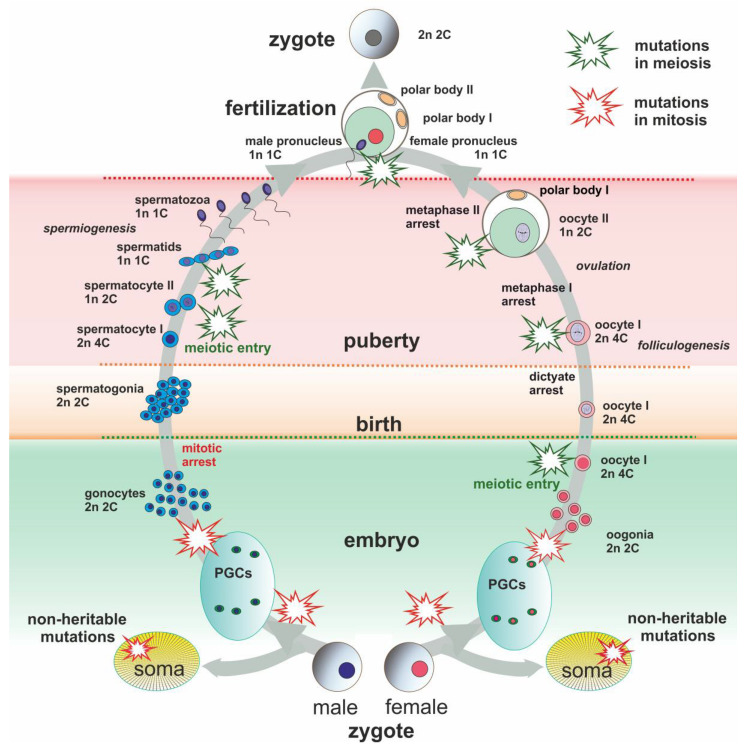
Schematic presentation of main events of male and female lifespan. Evolutionarily significant de novo chromosome rearrangements might occur during the lifespan and be transmitted to descendants. Somatic mutations are not inherited unless they can lead to the same type of chromosome rearrangements. Schematic explosions mark points of possible heritable mutations (green for meiotic origin and red for mitotic origin).

## Data Availability

Not applicable.

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
