# Peer review of "Chromosome Changes in Soma and Germ Line: Heritability and Evolutionary Outcome"

_genes, 2022, doi:10.3390/genes13040602_

Round 1
Reviewer 1 Report
Manuscript by author reviews an important and interesting topic. The author reviews the development of chromosomes during early development and the occurrence and evolution of mosaicism. Author also discusses the diversity of genome organization that is unique in different animals, including chromosomal changes in mammals and their internal relationships and factors that maintain genomes in natural evolution. Also the presentation of the article is simple and easy to read. Only line 147 “Dicentric chromosomes tend to vanish in several generations, possibly due to centromeric drive affecting two active centromeres”, is difficult to understand, If the author can explain the sentence more clearly.
Author Response
I am very grateful to the reviewers for their valuable comments, which help to improve the text and figures of the manuscript. Each comment has been carefully considered point by point and responded.
Only line 147 “Dicentric chromosomes tend to vanish in several generations, possibly due to centromeric drive affecting two active centromeres”, is difficult to understand, If the author can explain the sentence more clearly.
I have changed the sentence, and the meaning seems clearer now:
Even though different variants are formed, such chromosomes, probably due to the presence of two centromeres, are subjected to selection in the form of centromeric drive and are eliminated in the lineage within several generations.
Reviewer 2 Report
This manuscript aims to survey the different categories of heritable structural variants, the mechanisms by which they are generated, and the developmental stages at which those mechanisms operate. This is not my area of expertise, but the review appears to me to be scientifically sound, with appropriate references to the published literature. The writing style is largely grammatically correct, but the author uses mixed metaphors and awkward turns of phrase that are jarring and make the text difficult to read. It will require some editing to make the text clear and concise. The overall organization of the manuscript is my single largest concern. Presently, the manuscript seems closer in form to a catalogue than an essay. While the author is clearly aware of the many disparate lines of research that bear on the origins and genetic consequences of structural variants, it is difficult for me, as a non-specialist, to get a sense of which questions are most important, and which lines of experimentation will be most productive in answering them.
The introduction should be shortened and sharpened. It could be improved by posing the key questions the review aims to answer as early and as clearly as possible. Currently, the introduction meanders through several peripheral topics on the way to posing the central questions the review aims to answer. For example, about 15% of the introduction is spent on whether the genomic changes that occur in cancers can rightly be called evolution (for an extreme counterpoint, consider Tasmanian devil facial tumor disease), but the outcome of this mostly semantic debate has little relevance to the mechanistic and developmental origins of structural variants.
Later sections could be improved by increasing the focus around the central questions posed at the end of the introduction. I don't think the review ever actually addresses the question of "whether there are 'weak' patterns of genome organization that are more prone to chromosomal rearrangements and at which stages of individual development these changes are more likely to occur" (lines 97-99).
The author has mostly organized this review around the developmental trajectory of the mammalian germline (sections 2 and 3). Currently the manuscript surveys the literature to catalogue the various types of structural variants that emerge at each developmental stage, along with the mutational mechanisms that generate them, and some sense of the selective pressures that might cause these variants to be eliminated from the germ line. The manuscript could be strengthened by making the connections between mutational mechanisms and structural variants more explicit. Even more valuable would be a critical assessment of the literature to give readers a sense of which mechanisms and variants contribute the most to human disease or to evolution. Are some mechanisms or developmental stages more prone to generate variants of consequence than others? At a given developmental stage, are certain types of variants more likely to be passed to the next generation?
The framework abruptly shifts in the fourth section from following the development of the germline to discussing the 3D and 1D organization of the nucleus and various models of the origin of Robertsonian translocations. I found this transition somewhat confusing, and perhaps, if this review is intended to primarily highlight the causes and consequences of Robertsonian translocations, some of this material might be more appropriate for the introduction section. For example, it would make sense to highlight both the high frequency of these events in humans and other mammals (lines 297-301), and the consequences of these rearrangements (lines 267-274) at the very start of the manuscript. I was particularly interested in the various models of translocation origin in section 4.2, and would have preferred to see this better integrated with the developmental framework in the early part of the manuscript. Which of the (5?) competing models make testable predictions about the developmental origins of Robertsonian translocations? Which ones are falsified by the evidence in the literature? In biology, the answer to most "either ... or ..." questions is "yes" -- is it possible that each of the proposed mechanisms contribute to the origin of Robertsonian translocations? What is the best estimate of their relative contributions? What experiments or datasets are needed to refine this estimate?
The figures do not serve the text as well as they could. I offer a few comments on them below.
Figure 1 could be improved by illustrating either which types of structural variants arise at each "point of possible genomic rearrangement," or which mechanisms are in play at each point, or both. The evolutionary significance of the germline might be better illustrated by connecting this linear diagram into a circle (from zygote to zygote), with the soma branching off before the development of PGCs. This would accurately depict the origins of the soma, and make it clear that any mutations on the arrow leading to the soma are evolutionary dead ends.
Figure 2 is graphically redundant with figure 1; I would suggest deleting it from the manuscript and replacing it with an illustration of the several types of structural variants discussed in the introduction (lines 77-79, 84-85, 93-94), and identify the mechanisms that lead to each. Alternatively, illustrate the mechanisms and identify the rearrangements that result. It could be that a table would be more useful than a figure.
I don't understand what figure 3 is illustrating. Are heritability and "evolutionary outcome" really orthogonal dimensions? If they are, shouldn't there be some items that fall off the diagonal? Shouldn't the soma and germline differ along the heritability axis? Why are types of structural variants listed under mitosis, but mechanisms of structural variation listed under meiosis?
Author Response
I am very grateful to the reviewers for their valuable comments, which help to improve the text and figures of the manuscript. Each comment has been carefully considered point by point and responded.
- The introduction should be shortened and sharpened. It could be improved by posing the key questions the review aims to answer as early and as clearly as possible. Currently, the introduction meanders through several peripheral topics on the way to posing the central questions the review aims to answer. For example, about 15% of the introduction is spent on whether the genomic changes that occur in cancers can rightly be called evolution (for an extreme counterpoint, consider Tasmanian devil facial tumor disease), but the outcome of this mostly semantic debate has little relevance to the mechanistic and developmental origins of structural variants.
Thank you very much for your comments. The comments greatly improved the consistency of the text and, I hope, made it easier to understand. I have reduced the Introduction and removed some of the speculation to the last section of the paper. I am also grateful for the suggestion to refer to an example of Tasmanian devil facial-tumour disease, which perfectly illustrates the evolutionary dead end in the case of the cancer. Following text was added:
The example of the Tasmanian devil, suffering from the facial-tumour disease, in which massive changes in chromosomes were observed, became the cause of extinction, confirmed the point [112 Pearse, A.M. and Swift, K., 2006. Transmission of devil facial-tumour disease. Nature, 439(7076), pp.549-549.]
- Later sections could be improved by increasing the focus around the central questions posed at the end of the introduction. I don't think the review ever actually addresses the question of "whether there are 'weak' patterns of genome organization that are more prone to chromosomal rearrangements and at which stages of individual development these changes are more likely to occur" (lines 97-99).
I am grateful for the comment, this paragraph has been deleted, the text of the Introduction was re-structured. I believe that centromeric and telomeric regions are probably those major chromosomal regions that undergo global-level rearrangements more frequently. The accent on the problem was made in other parts of the ms.
- The author has mostly organized this review around the developmental trajectory of the mammalian germline (sections 2 and 3). Currently the manuscript surveys the literature to catalogue the various types of structural variants that emerge at each developmental stage, along with the mutational mechanisms that generate them, and some sense of the selective pressures that might cause these variants to be eliminated from the germ line. The manuscript could be strengthened by making the connections between mutational mechanisms and structural variants more explicit. Even more valuable would be a critical assessment of the literature to give readers a sense of which mechanisms and variants contribute the most to human disease or to evolution. Are some mechanisms or developmental stages more prone to generate variants of consequence than others? At a given developmental stage, are certain types of variants more likely to be passed to the next generation?
Many thanks for the constructive comments. In the subsections I have tried to focus specifically on the contribution of individual rearrangements, to evaluate the evolutionary consequences. Molecular mechanisms of rearrangements are currently practically unstudied. We can only rely on single works on a few model objects.
- The framework abruptly shifts in the fourth section from following the development of the germline to discussing the 3D and 1D organization of the nucleus and various models of the origin of Robertsonian translocations. I found this transition somewhat confusing, and perhaps, if this review is intended to primarily highlight the causes and consequences of Robertsonian translocations, some of this material might be more appropriate for the introduction section. For example, it would make sense to highlight both the high frequency of these events in humans and other mammals (lines 297-301), and the consequences of these rearrangements (lines 267-274) at the very start of the manuscript. I was particularly interested in the various models of translocation origin in section 4.2, and would have preferred to see this better integrated with the developmental framework in the early part of the manuscript. Which of the (5?) competing models make testable predictions about the developmental origins of Robertsonian translocations? Which ones are falsified by the evidence in the literature? In biology, the answer to most "either ... or ..." questions is "yes" -- is it possible that each of the proposed mechanisms contribute to the origin of Robertsonian translocations? What is the best estimate of their relative contributions? What experiments or datasets are needed to refine this estimate?
I changed the structure of the review in an effort to follow the reviewer's recommendations. The part on the significance of the chromosome arrangement in the nucleus and the models of translocation formation has been moved to the beginning, section 2. The question of chromosome speciation has been left at the end, before that, arguments about the gradual or sudden nature of evolution have been inserted.
- The figures do not serve the text as well as they could. I offer a few comments on them below.
Figure 1 could be improved by illustrating either which types of structural variants arise at each "point of possible genomic rearrangement," or which mechanisms are in play at each point, or both. The evolutionary significance of the germline might be better illustrated by connecting this linear diagram into a circle (from zygote to zygote), with the soma branching off before the development of PGCs. This would accurately depict the origins of the soma, and make it clear that any mutations on the arrow leading to the soma are evolutionary dead ends.
Figure 2 is graphically redundant with figure 1; I would suggest deleting it from the manuscript and replacing it with an illustration of the several types of structural variants discussed in the introduction (lines 77-79, 84-85, 93-94), and identify the mechanisms that lead to each. Alternatively, illustrate the mechanisms and identify the rearrangements that result. It could be that a table would be more useful than a figure.
I don't understand what figure 3 is illustrating. Are heritability and "evolutionary outcome" really orthogonal dimensions? If they are, shouldn't there be some items that fall off the diagonal? Shouldn't the soma and germline differ along the heritability axis? Why are types of structural variants listed under mitosis, but mechanisms of structural variation listed under meiosis?
Figure 1 has been radically changed, transformed into a cyclic scheme and supplemented with indications of the types of rearrangements. Figures 2 and 3 have been deleted. A table summarizing the information about the possible place and time of occurrence of rearrangements has been inserted.
